# Predicting learning achievement using ensemble learning with result explanation

**Tingting Tong**◉*, **Zhen Li**

School of Information Science and Technology, Northeast Normal University, Changchun, Jilin, China

* tongtt616@nenu.edu.cn

## Abstract

Predicting learning achievement is a crucial strategy to address high dropout rates. However, existing prediction models often exhibit biases, limiting their accuracy. Moreover, the lack of interpretability in current machine learning methods restricts their practical application in education. To overcome these challenges, this research combines the strengths of various machine learning algorithms to design a robust model that performs well across multiple metrics, and uses interpretability analysis to elucidate the prediction results. This study introduces a predictive framework for learning achievement based on ensemble learning techniques. Specifically, six distinct machine learning models are utilized to establish a base learner, with logistic regression serving as the meta learner to construct an ensemble model for predicting learning achievement. The SHapley Additive exPlanation (SHAP) model is then employed to explain the prediction results. Through the experiments on XuetangX dataset, the effectiveness of the proposed model is verified. The proposed model outperforms traditional machine learning and deep learning model in terms of prediction accuracy. The results demonstrate that the ensemble learning-based predictive framework significantly outperforms traditional machine learning methods. Through feature importance analysis, the SHAP method enhances model interpretability and improves the reliability of the prediction results, enabling more personalized interventions to support students.

## Introduction

In recent years, Massive Open Online Courses (MOOCs) have gained global popularity for providing free, high-quality learning resources and enhanced support for students [1]. However, despite high enrollment rates, the persistent challenges of high dropout rates and low engagement during the learning process remain significant [2].

Machine learning (ML) technologies have emerged as promising tools to tackle student attrition by predicting learning achievement [3, 4]. Numerous studies have focused on developing ML algorithms for this purpose [5–9]. However, these algorithms face challenges in feature processing and optimal algorithm selection due to varying data perspectives and objectives in data mining [10, 11]. The predictive accuracy of these algorithms often suffers because optimal hyper-parameter settings depend heavily on dataset characteristics, necessitating customized configurations for optimal performance [12].

**Data Availability Statement:** All relevant data for this study are publicly available from the Zenodo repository (https://zenodo.org/records/13892715).

**Funding:** This study was financially supported by Natural Science Foundation of Jilin Province in the form of a grant (YDZJ202201ZYTS421) received

by ZL. This study was also financially supported by National Natural Science Foundation of China in the form of a grant (62007005) received by ZL. This study was also financially supported by Fundamental Research Funds for the Central Universities in the form of a grant (2412022ZD017) received by ZL.

**Competing interests:** The authors have declared that no competing interests exist.

Predicting potential academic vulnerabilities among students using artificial intelligence frameworks is crucial for devising targeted interventions. Moreover, understanding the predictive models' explanations provides valuable insights into the underlying reasons for students' vulnerabilities, facilitating personalized interventions tailored to their specific needs [13].

This study tackles the crucial task of identifying students at risk of dropping out and proposes targeted interventions to mitigate their academic challenges. It advocates two main strategies: first, employing an ensemble learning approach that leverages the strengths of diverse predictive models to enhance the accuracy of learning achievement predictions; and second, utilizing model-agnostic explanatory techniques to pinpoint specific student features associated with academic risks.

Specifically, this research proposes a prediction model based on stacking ensemble learning. Stacking ensemble models typically employ heterogeneous learners to develop multiple base models concurrently, followed by the construction of meta learner to aggregate the final prediction outcomes [14].

The proposed model is rigorously trained and verified through experimental methodologies and comprehensive outcome analyses to demonstrate its effectiveness in accurately predicting learning outcomes. Furthermore, the study introduces a model-agnostic technique utilizing the SHAP (SHapley Additive exPlanations) [15], an innovative method to interpretability analysis. This method is anticipated to provide new insights into pedagogical interventions by offering a deeper understanding of the model's predictions.

To summarize, the main contributions and novelty of our work are as follows:

- We developed a robust ensemble learning model that integrates six distinct machine learning models (K-Nearest Neighbor, Naive Bayes, Random Forest, Gradient Boosting Decision Tree, eXtreme Gradient Boosting, and Multi-Layer Perceptron) as base learners, with Logistic Regression as the meta learner. This model effectively addresses the biases and limitations of previous methodologies.

- To enhance the interpretability of our predictions, we employed the SHapley Additive exPlanation (SHAP) for feature importance analysis. This allowed us to identify critical factors influencing learning achievement, providing actionable insights for more precise and targeted interventions.

- The effectiveness of the proposed model was verified through experiments on the XuetangX dataset. The proposed model outperforms traditional machine learning and deep learning methods in terms of prediction accuracy.

The remainder of this paper is structured as follows: The Related work section reviews previous studies and the foundation for this research. The Ensemble learning achievement prediction and explanation of prediction results section outlines the overall research framework and provides details of the proposed model structure. The Experimental design and result analysis section presents the setup, experiments conducted, and a detailed analysis of the results obtained. The Conclusions and limitations section summarizes the key findings of the study and discusses the limitations and potential areas for future research.

## Related work

### Learning achievement prediction

**Predictive features.** Predicting learning achievement involves leveraging various learner features, which can be categorized into demographic information and online behavioral data.

Demographic information typically includes factors such as gender, age, and educational background, while online behavioral data encompasses metrics like video consumption, time spent on course materials, and participation in online activities.

Studies such as [16] have leveraged demographic factors like age, gender, and prior academic performance to forecast learning outcomes. These approaches employed interpretable machine learning techniques to identify factors contributing to poor performance and integrated rule-based risk models to enhance prediction accuracy.

Another study [17] focused on basic learner information to predict learning achievement. This research presented a two-stage predictive model development process aimed at improving recognition accuracy and supporting educators in implementing diverse teaching practices to enhance student learning outcomes.

Additionally, research by [18] incorporated features like gender, age, and residential status into a nonlinear State Space Model tailored for predicting student dropout. This model emphasized the evolving latent state of students in open and distance education settings, highlighting the importance of ongoing student status monitoring.

However, these studies often rely on static input features that do not account for dynamic learning processes, serving more as early warning systems prior to actual learning engagement. They often fail to capture the behaviors of learners during their learning activity, thereby limiting prediction accuracy.

In contrast, online interaction data emerged as a critical predictor of learning achievement, offering insights into individual learning quality [19, 20]. For instance, [21] used Logistic Regression (LR) to extract features from learners' interactions with video lectures and assignments, predicting learning performance based on these behaviors. Similarly, [22] employed Random Forest (RF) to model learning achievements using online clickstream data.

Most studies within the MOOCs context have prioritized behavioral data for predicting learning outcomes [23, 24]. However, both behavioral data and demographic information are crucial predictors of learning achievement. Therefore, this paper integrates demographic information and online behavioral data to predict learning achievement.

**Prediction models.** The development of learning achievement prediction models involves a diverse range of machine learning methods extensively studied by researchers. These models can be categorized into two primary groups: traditional machine learning algorithms and ensemble learning methods.

*Traditional machine learning algorithms.* Marbouti [25] employed LR, Decision Trees (DT), Naive Bayes (NB), K-Nearest Neighbor (KNN), Artificial Neural Networks (ANN), and Support Vector Machines (SVM) to develop a robust model aimed at identifying learners at risk of failure. Their study highlighted the challenge of accurately identifying both successful and unsuccessful learners across different algorithms.

Howard [26] systematically compared the performance of common prediction algorithms, including RF, SVM, ANN, and KNN, among others. Their findings indicated that RF yielded the most effective results.

In another investigation [27], DT, LR, NB, and RF algorithms were evaluated to recommend an optimal choice for predicting dropout. The study concluded that straightforward algorithms could achieve reliable accuracy in identifying predictors of dropout.

Similarly, a study [28] utilized RF, SVM, DT, NB, KNN, and LR for classification tasks, with RF demonstrating superior performance in detecting students susceptible to dropout, achieving an accuracy rate of 94.14%.

These studies underscore the efficacy of machine learning in addressing student learning achievement prediction. However, the choice of prediction algorithm depends on the

perspective of data observation, mining objectives, and research context, making it challenging to determine a universally superior algorithm for online learning achievement prediction.

The objective of employing machine learning technology in building learning achievement prediction models is to achieve high accuracy, strong generalization ability, and robustness through rigorous training. Nonetheless, in practical applications, machine learning models often exhibit biases that hinder them from fully meeting operational requirements.

*Ensemble learning methods.* Ensemble learning stands as a critical approach in machine learning, combining multiple weak learners to form a robust model with improved accuracy and generalization capabilities [29, 30]. Methods like Bagging, Boosting, and Stacking are particularly effective in improving model performance and addressing both classification and regression tasks.

Bagging and Boosting are homogeneous ensemble methods, which rely on using the same base learning algorithm across multiple iterations. Conversely, Stacking is a heterogeneous ensemble approach that employs diverse base learners in parallel, combining their outputs through a meta learner to generate the final prediction [31]. This approach increases model diversity and enhances generalization, offering distinct advantages over homogeneous ensemble methods.

Ensemble learning algorithms have broad applications across various fields. For example, recent models like Deepstacked-AVPs, iAFPs-Mv-BiTCN, pAVP_PSSMDWT-EnC, iACP-- GAEnsC, and CACP, developed by Akbar et al. [32–36], significantly enhance peptide identification and prediction by integrating advanced feature selection techniques and optimized algorithms. These models have proven to be highly valuable in pharmaceutical design and research. Similarly, Ullah et al. [37] developed the DeepAVP-TPPred model, which improves antiviral peptide prediction using a novel binary tree growth algorithm.

These examples highlight the wide-ranging applicability and impact of ensemble learning methods across different domains, demonstrating their effectiveness in addressing complex problems through model integration and optimization.

**Data preprocessing.** Data preprocessing is a critical step in building predictive models for learning achievement, involving tasks such as data cleaning and transformation. This section discusses two key aspects of data preprocessing: addressing class imbalance and performing data transformation.

In learning achievement prediction research, student grades are central indicators of academic performance and serve as targets for both regression and classification tasks. However, datasets used for modeling often exhibit class imbalance, with students achieving extremely low or high grades representing only a small portion of the overall data. The class imbalance issue can severely affect the predictive performance of the learning achievement prediction.

Many studies in the field proceed with modeling based on imbalanced datasets without directly addressing this issue. For example, Al-Musharraf et al. [38] categorized course grades into five classes for learning achievement prediction but observed a disproportionately small number of students in the highest performing class (Class A). Only a few researchers have explored resampling strategies to mitigate the impact of data imbalance on predictive models. For instance, Romero et al. [39] applied random oversampling to rebalance their dataset and assessed the performance of predictive models before and after resampling. Their findings revealed that while resampling improved the performance of some algorithms, its effects varied across different models.

Therefore, identifying the optimal sampling strategy to enhance the effectiveness of predictive models, particularly in the context of the unique characteristics of educational data, remains an area that requires further research and exploration.

## Model interpretability analysis

In balancing the trade-off between predictive accuracy and interpretability, previous research has primarily focused on enhancing interpretability by identifying significant predictors using traditional statistical methods. For instance, one study [40] emphasized the importance of student attributes in predicting academic success through variable importance analysis, highlighting that active participation in forums during video lectures positively correlates with course success. Similarly, another study [41] employed Bayesian algorithms to identify age and scholarship as crucial predictors of learning achievement.

While these studies offer valuable insights into key features, they often fall short in explaining how these features specifically contribute to predictions, underscoring the need for further advancements in model interpretability.

To bridge this gap, Lundberg [15] introduced SHAP, a comprehensive framework designed to enhance the interpretability of machine learning models. SHAP calculates linear additive contributions for each feature variable across samples, providing detailed explanations. Unlike traditional feature importance analyses, SHAP offers both global and local interpretability. Globally, it ranks feature importance, identifies key predictors influencing predictions, and assesses the qualitative impact of features on outcomes. Locally, SHAP elucidates the specific role of each feature in predicting outcomes for individual samples, significantly enhancing the reliability of predictions.

In this study, we integrate ensemble learning algorithms with the interpretable machine learning framework offered by SHAP to construct a predictive model for learning achievement. This combined approach not only improves predictive accuracy but also provides robust interpretability, making it highly effective for identifying nuanced factors that influence learning outcomes.

## Research question

Building on an extensive review of existing literature, this study aims to develop an ensemble learning strategy for predicting learning achievement, with a strong emphasis on interpretability. To achieve this goal, the research is organized around two primary sub-questions:

Research Question 1: How can an ensemble learning framework be designed to accurately predict learning achievement?

Research Question 2: How can the results of ensemble learning predictions be effectively interpreted using the SHAP method?

By addressing these sub-questions, this research seeks not only to enhance the accuracy of learning achievement predictions through ensemble learning but also to provide comprehensive interpretability of model outcomes using advanced machine learning techniques. This dual focus is essential for identifying critical factors influencing learning achievement and enabling informed interventions in educational settings.

## Ensemble learning achievement prediction and explanation of prediction results

### Research framework

This study proposes an ensemble learning approach to predict learning achievement by utilizing behavioral data from learning activities and student demographic information. It employs six independent machine learning models—KNN, NB, RF, GBDT, XGBoost, and MLP—as base learners, with Logistic Regression (LR) serving as the meta learner to construct a stacking ensemble model. The research methodology includes data analysis, model training using

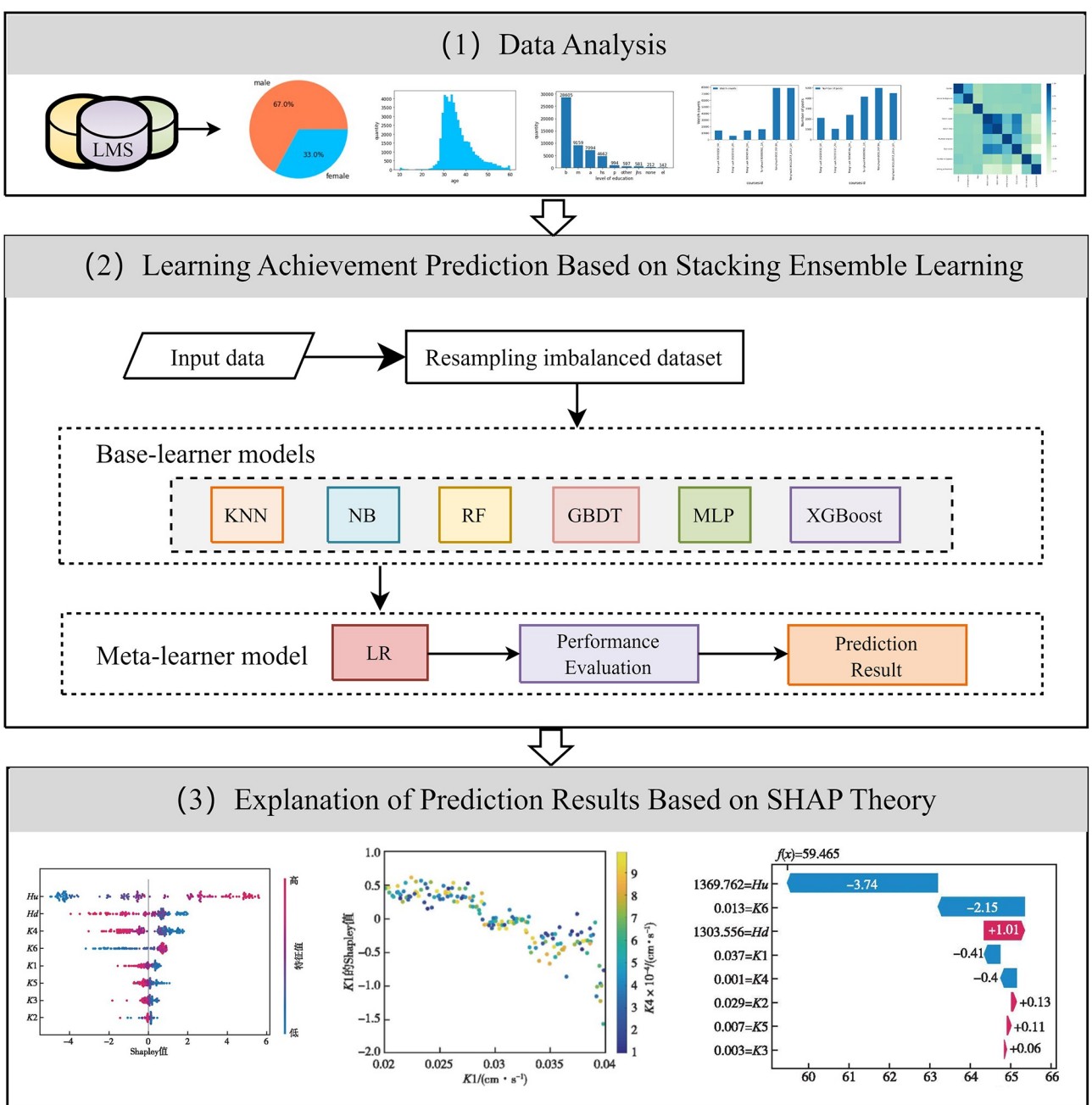

**Fig 1. Research framework.**

learner profiles and interaction data, evaluation of prediction accuracy against baseline models, and interpretation of results using the SHAP method, as illustrated in Fig 1. This framework aims to enhance prediction accuracy while providing insights into the factors influencing learning outcomes, thereby facilitating targeted educational interventions.

**Data analysis.** *Research context and participants*. This study utilized data sourced from XuetangX (https://www.XuetangX.com), encompassing 59,581 learners across six courses: Circuit Principles (I) (courseid: TsinghuaX-20220332_1X-_), Circuit Principles (II) (courseid:

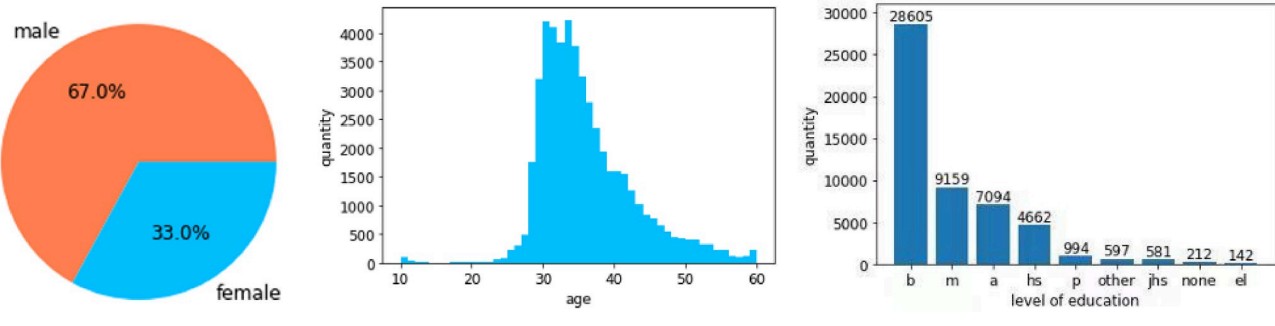

**Fig 2. Statistics of participants.**

TsinghuaX-20220332_2X-_), Data Structures (courseid: TsinghuaX-30240184_1X-_), History of Chinese Architecture (courseid: TsinghuaX/80000901_1X_), Financial Analysis and Decision (Fall 2013) (courseid: TsinghuaX-80512073X-_), and Financial Analysis and Decision (Spring 2014) (courseid: TsinghuaX-80512073_2014_1X-_). Four of these courses were conducted from October 10, 2013, to January 2014, and two from March to June 2014, each spanning a duration of 10 weeks.

A demographic analysis of the learners, depicted in Fig 2, revealed that 67% of participants were male and 33% were female, with ages primarily ranging between 20 and 50 years old. The educational background distribution included 28,605 learners with a bachelor's degree, 9,159 with a master's degree, 7,094 with an associate's degree, 4,662 with a high school diploma, and 994 with a doctoral degree. The data utilized in this study comprised both learner demographic information and behavioral data collected during the courses.

A comprehensive overview of the learner data is summarized in Table 1. For the prediction phase, data from all 59,581 learners were utilized, among whom 3,155 learners obtained a pass certificate, while the remaining 56,426 learners did not receive a certificate.

In this study, the total MOOC score ranges from 0 to 100. Learners who score between 60 and 100 receive a passing certificate, while those scoring below 60 do not receive a certificate. The relationship between learning achievement and certificate attainment is shown in Table 2. Among the learners in this dataset, 3,155 (5.29%) received a passing certificate, while 56,426 (94.77%) did not. The certificate attainment rate of 5.29% is consistent with typical MOOC patterns, which generally range between 3.5% and 7.3%.

*Feature histogram.* Regarding the learning behavioral data, histograms were generated to depict the frequency distributions of watch counts and the number of posts. These histograms

**Table 1. Details of the dataset.**

| Leaner data | Feature | Description |
|---|---|---|
| Demographic information | Gender | The learner's gender. |
| | Age | The learner's age. |
| | Educational background | The learner's level of education. |
| Behavioral data | Watch count | The number of videos watched by learners. |
| | Watch time | Duration of video watched by learners. |
| | Number of posts | The number of posts learners have made on the forum. |
| | Number of quizzes | The number of times the learner takes the quiz. |
| Academic performance | Quiz score | Learner's scores on assignments and quizzes. |

**Table 2. Learning achievement.**

| Learning achievement | n | % |
|---|---|---|
| Pass certificate | 3155 | 5.29% |
| Non-certificate | 56426 | 94.77% |
| All learners | 59581 | 100% |

specifically illustrate the frequency of video clicks and forum postings by learners in each course, as shown in Fig 3. From these figures, it is evident that learners in certain courses, such as History of Chinese Architecture (courseid: TsinghuaX/80000901_1X_), Financial Analysis and Decision (Spring 2014) (courseid: TsinghuaX-80512073_2014_1X-_), and Financial Analysis and Decision (Fall 2013) (courseid: TsinghuaX-80512073X-_), demonstrate higher levels of engagement in forum activities. This observation underscores their active participation in discussions within these courses.

*Correlation analysis*. Correlation analysis uses statistical indicators to quantify the degree of linear association between continuous variables. Common methods include creating scatter plots, constructing scatter plot matrices, and calculating correlation coefficients. In bivariate correlation analysis, the Pearson correlation coefficient, Spearman correlation coefficient, and Kendall's tau coefficient are commonly used. This paper utilizes the Pearson correlation coefficient to evaluate the strength of the relationships between dependent and independent variables. The Pearson correlation coefficient is given by Eq (1):

$$r = \frac{\sum_{i=1}^{n}(x_i - \bar{x})(y_i - \bar{y})}{\sqrt{\sum_{i=1}^{n}(x_i - \bar{x})^2 \sum_{i=1}^{n}(y_i - \bar{y})^2}} \tag{1}$$

The Pearson correlation coefficient was employed to analyze the raw data, which included eight independent variables and one dependent variable related to learning achievement

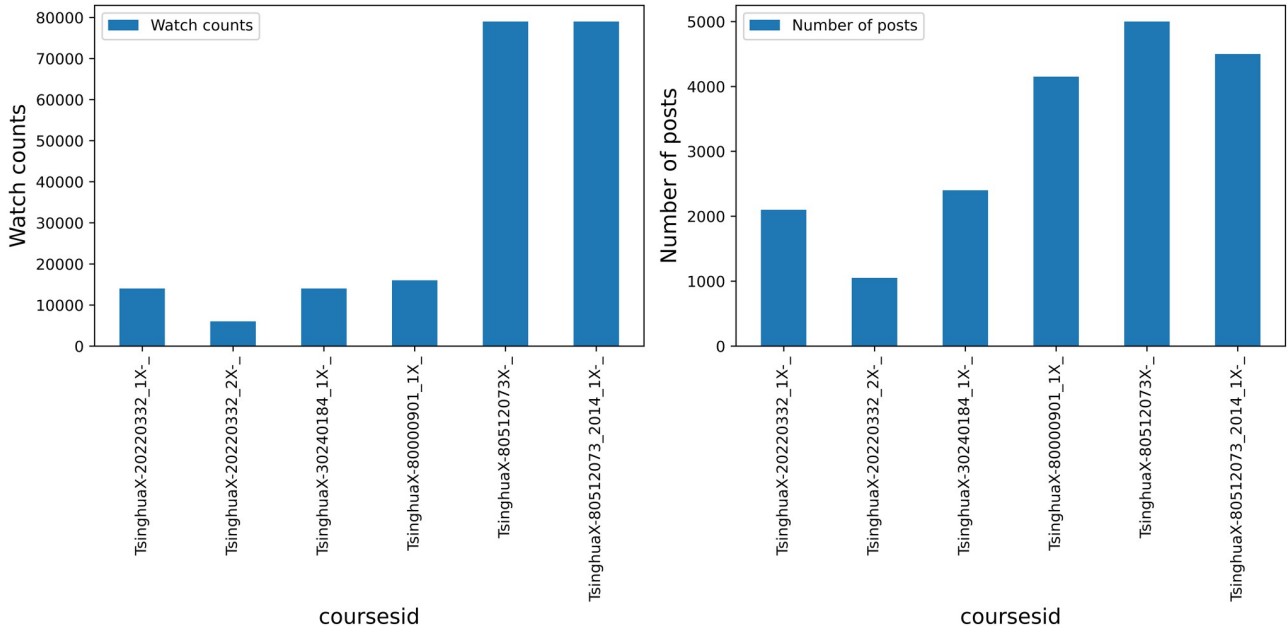

**Fig 3. Feature frequency distribution histogram.**

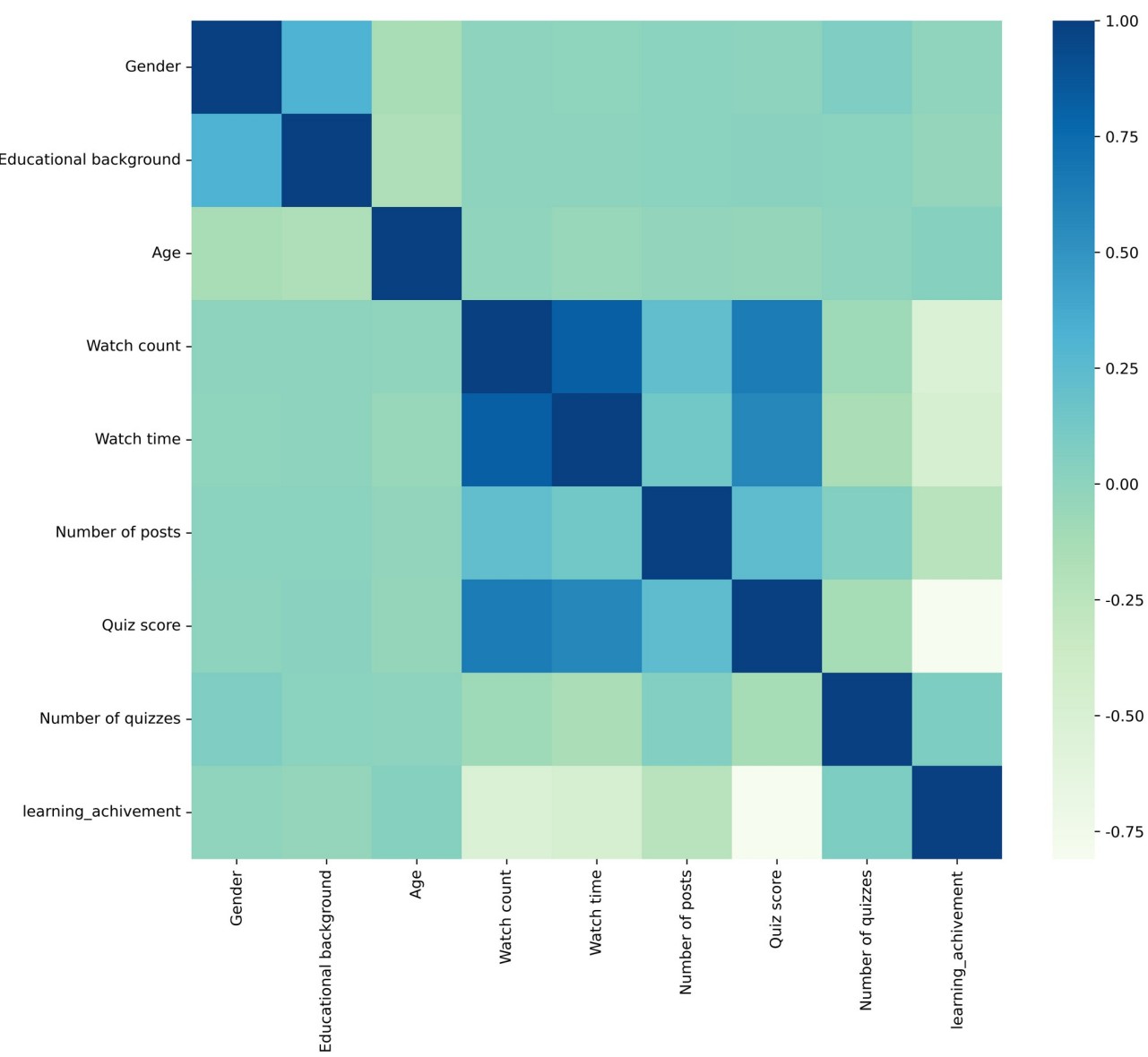

**Fig 4. Pearson correlation heatmap.**

prediction. To visually represent the degree of correlation between these variables, a heatmap of the correlation coefficient matrix was generated using Python. This heatmap uses color intensity to illustrate the strength of correlations, as shown in Fig 4.

*Data resampling.* Addressing data imbalance is a common challenge in classification tasks, particularly with real-world datasets like those from online learning environments. This issue often leads to skewed distributions, where one class (e.g., dropout students) is significantly underrepresented compared to another (e.g., students who complete their studies). Such imbalances can distort predictive models, causing them to disproportionately favor the dominant class while neglecting instances of the minority class.

Traditional evaluation metrics like accuracy can be misleading in imbalanced datasets because high accuracy may mask the model's inability to effectively predict the minority class.

In dropout prediction tasks, for instance, where the number of students completing their studies is relatively small compared to those who drop out, models may incorrectly favor predicting dropout, thereby compromising overall performance and generalization ability.

To address these challenges, various resampling techniques are employed [42, 43]. Oversampling involves increasing the number of minority class samples by duplicating or synthetically generating new ones, with methods like SMOTE (Synthetic Minority Over-sampling Technique) [44] and ADASYN (Adaptive Synthetic Sampling Approach) [45] being particularly popular. In contrast, undersampling involves randomly removing samples from the majority class to balance the dataset, with techniques like Tomek links and NearMiss commonly used [46]. Mixed sampling combines both oversampling and undersampling to balance class distributions more effectively [47].

Despite the availability of these resampling methods, their application and effectiveness in educational contexts, particularly in predicting student dropout, remain underexplored. It is essential to determine which resampling approach is best suited for educational datasets characterized by class imbalance. This study aims to fill these gaps by evaluating various resampling techniques to enhance the predictive performance of dropout prediction models in academic settings.

**Learning achievement prediction based on stacking ensemble learning.** Given the intricate relationships within educational data and the unique strengths of different algorithms, this study employs ensemble learning techniques to improve the accuracy of learning achievement predictions. Specifically, we utilize the stacking ensemble approach, which combines multiple algorithms to enhance overall model performance.

The stacking algorithm uses a hierarchical blending strategy, where various base learners are integrated through a meta learner to boost model accuracy. To reduce overfitting, we select logistic regression as the meta learner. The stacking model consists of two layers: the first layer includes heterogeneous base learners, and the second layer involves the meta learner. The training set is divided using k-fold cross-validation (CV), where each base learner's predictions are used as inputs for the meta learner, ultimately leading to the final prediction.

Unlike homogeneous ensemble methods that rely on similar base learners, stacking uses diverse learners in parallel, enhancing model diversity and generalization. This makes it particularly effective for predicting learning outcomes in varied educational settings.

Originally introduced by David H. Wolpert in 1992 [48], stacking differs from Bagging and Boosting by combining the outputs of diverse base learners through a meta learner rather than using identical base models. This method, illustrated in Fig 5, enhances model robustness and flexibility, making it a powerful tool in ensemble learning. The process involves splitting the training set into k folds, training base learners on k-1 folds, and using these predictions to train the meta learner. For this study, k was set to 5. The implementation of a two-layer stacking framework involves the following four steps:

- Divide the Training Set: Split the training set into 5 folds using 5-fold cross-validation.

- Train Base Learners: Train the base learners on 4 folds and predict the 5-th fold. Repeat this process for each fold, then concatenate the predictions to form a new training set while retaining the original labels.

- Train Meta Learner: Train the meta learner on the new training set constructed from the base learners' predictions.

- Predict New Test Set: Use the trained meta learner to predict the new test set samples, yielding the final predictions.

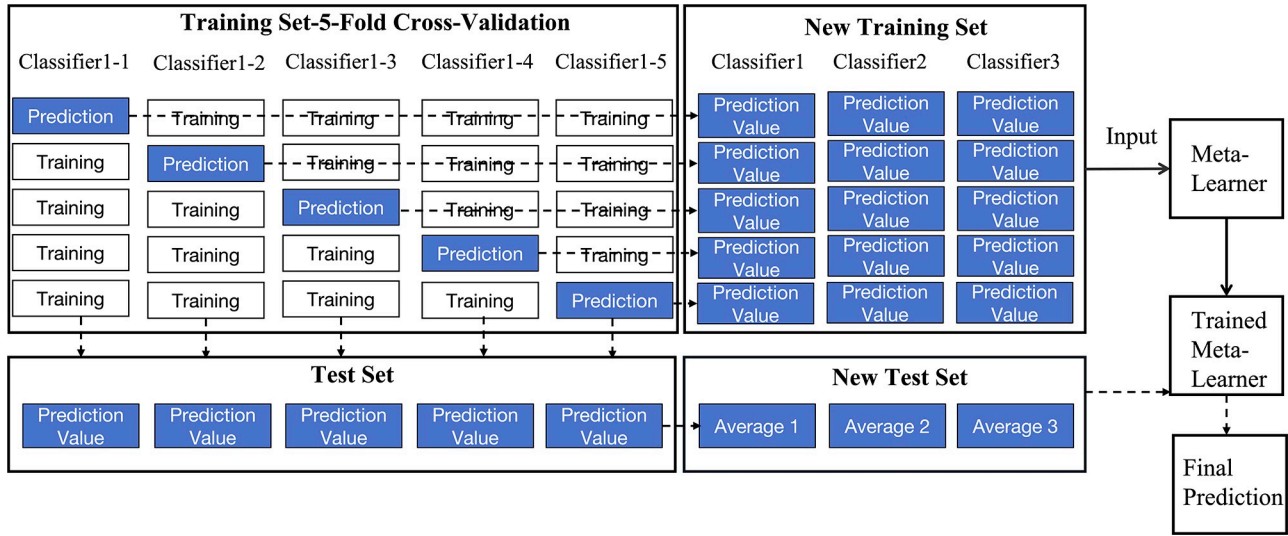

**Fig 5. Flowchart of the stacking method.**

**Explanation of prediction results based on SHAP theory.** In the realm of predicting learning achievement, understanding the inner workings of machine learning algorithms is crucial for meaningful interpretation of their predictions. Existing algorithms, often perceived as "black boxes", require interpretability analysis to clarify how they arrive at their predictions [49]. This is particularly important in educational contexts, where transparency and insights into prediction outcomes are essential for developing effective intervention strategies.

While traditional ensemble learning models excel in ranking feature importance, they often lack in providing detailed insights into how each feature contributes to individual prediction outcomes. To address this limitation, this study integrates SHAP theory with ensemble learning algorithms. SHAP theory enables comprehensive global importance analysis of features, identifying key predictors that significantly influence learning achievement predictions.

Beyond merely ranking feature importance, SHAP theory elucidates the directional impact of input features on prediction outcomes. It quantifies both positive and negative correlations between features and prediction results, offering nuanced insights into the interactions among features and their respective influences on learning achievement predictions. This analytical approach not only enhances the reliability of the prediction model but also provides novel perspectives for designing targeted teaching interventions tailored to individual student needs.

By integrating SHAP theory with ensemble learning, this study aims to bridge the gap between predictive accuracy and interpretability, thereby empowering educators and researchers with actionable insights to foster student success in educational settings.

## Learning achievement prediction using ensemble learning

**The proposed model.** To enhance predictive accuracy in learning achievement prediction, this study employs ensemble learning by integrating multiple machine learning algorithms through a stacking approach. Six diverse algorithms have been selected: K-Nearest Neighbors (KNN), Naive Bayes (NB), Random Forest (RF), Gradient Boosting Decision Tree (GBDT), eXtreme Gradient Boosting (XGBoost), and Multi-Layer Perceptron (MLP). These algorithms were chosen for their effectiveness in handling classification tasks and their ability

to complement each other's strengths within an ensemble framework. Below is a brief overview of each algorithm:

- K-Nearest Neighbors (KNN): KNN is a simple yet effective classification algorithm that assigns a new data point to the most common category among its K nearest neighbors, determined by Euclidean distance.

- Naive Bayes (NB): NB is a probabilistic classifier based on Bayes' theorem and assumes conditional independence among features. It calculates the posterior probability of each class given the input features and predicts the class with the highest probability.

- Random Forest (RF): RF is a Bagging ensemble learning method that constructs multiple decision trees and aggregates their predictions through voting. It reduces overfitting by randomly selecting features and samples during tree construction.

- Gradient Boosting Decision Tree (GBDT): GBDT builds decision trees sequentially, with each tree correcting the errors of its predecessor. It combines the strengths of boosting and decision trees, achieving high accuracy but requiring careful parameter tuning.

- eXtreme Gradient Boosting (XGBoost): XGBoost is an optimized implementation of gradient boosting that enhances performance and computational speed. It uses a more regularized model to control overfitting and is known for its efficiency in handling large datasets.

- Multi-Layer Perceptron (MLP): MLP is a type of neural network consisting of multiple layers, including input, hidden, and output layers. It learns complex patterns in data through forward propagation and backpropagation of errors, requiring substantial computational resources and data.

The stacking ensemble learning approach integrates diverse algorithms into a hierarchical framework. In the first layer, each base learner (KNN, NB, RF, GBDT, XGBoost, MLP) independently processes the input data and generates predictions. These predictions are then passed to the second layer, where a meta learner (LR in this study) aggregates them to produce the final prediction. The pseudocode for the proposed model is outlined below.

**Algorithm 1** Pseudocode of the Stacking Ensemble Learning Model

```
Require: Training set D = {(x₁, y₁), (x₂, y₂), ..., (xₙ, yₙ)}; Base
learning algorithms 𝓛₁,𝓛₂,...,𝓛_𝒯; Meta learning algorithm 𝓛;
Ensure: Trained ensemble model H(x)
  1: Phase 1: Training Base Learners
  2: for t = 1, 2, ..., T do
  3:   Train the t-th base learner hₜ on the full dataset D:
  4:     hₜ = 𝓛ₜ(D);
  5: end for
  6: Phase 2: Generating Meta-Features
  7: Initialize the meta-training set D' = ∅;
  8: for i = 1, 2, ..., n do          ▷ For each training instance
  9:   Initialize meta-feature vector zᵢ = ∅;
 10:    for t = 1, 2, ..., T do          ▷ For each base learner
 11:      Compute prediction zᵢₜ = hₜ(xᵢ);
 12:      Append zᵢₜ to zᵢ;
 13:    end for
 14:   Add the meta-feature vector and the true label to meta-training
       set:
 15:     D' = D' ∪ (zᵢ, yᵢ);
 16: end for
 17: Phase 3: Training Meta Learner
 18: Train the meta-learner h' on the meta-training set D':
```

```
19: h' = L(D');
20: Phase 4: Making Predictions with the Ensemble Model
21: Define the final ensemble model as:
22: return H(x) = h'(h₁(x), h₂(x), ..., h_T(x)).
```

## Explanation of prediction results based on SHAP

In the practical deployment of machine learning models, achieving high predictive accuracy is just the first step. Equally important is understanding why a model makes specific predictions, as this insight is essential for refining the model's effectiveness and gaining a deeper understanding of its operational logic. Such interpretability not only enhances the reliability of the model but also supports educators and system managers in making informed decisions based on predictive outcomes.

In the context of learning achievement prediction, interpretability goes beyond merely identifying important features. It involves clarifying the extent of their impact and how these features influence the model's decision-making process. This level of interpretability is crucial for stakeholders who need actionable insights from predictive models in educational settings.

Explainable Artificial Intelligence (XAI) has emerged as a key area of research, focusing on developing machine learning models that are not only accurate but also transparent and interpretable. The SHAP framework, introduced by Lundberg [15], addresses this need by offering a unified approach to enhance model explainability.

Traditional machine learning algorithms often evaluate feature importance to identify key predictors influencing outcomes. However, they are typically deficient in explaining how these features precisely impact predictions. In contrast, SHAP provides a more comprehensive approach: it ranks feature importance, identifies critical predictors, and quantitatively analyzes their positive and negative correlations with prediction outcomes. Additionally, SHAP offers insights into how each feature of a specific sample contributes to the final prediction, thereby significantly improving the reliability and interpretability of model predictions.

SHAP accomplishes this by calculating the Shapley value for each feature, which measures its impact on the model's output. This methodological approach not only enhances the understanding of complex machine learning models but also fosters trust and acceptance among users by making the decision-making process more transparent and accessible.

In summary, integrating SHAP into learning achievement prediction models not only boosts their predictive capabilities but also provides stakeholders with clear insights into the factors driving educational outcomes. This enables more informed decision-making and supports the design of targeted interventions to effectively improve learning outcomes. The Shapley value of each feature is calculated as shown in Eq (2):

$$\varphi_i = \sum_{S \subseteq F \setminus \{i\}} \frac{|S|!(|F| - |S| - 1)!}{|F|!} f_{s \cup \{i\}}(\chi_{s \cup \{i\}}) - f_s(\chi_s) \tag{2}$$

Where, $S$ is the feature subset used in the model; $F$ represents the set of all features; $f_{s \cup \{i\}}(\chi_{s \cup \{i\}})$ represents the model output value for input features $i$ and feature subsets $S$; $f_s(\chi_s)$ represents the model output value when only a subset of features $S$ as input.

## Experimental design and result analysis

This section presents and discusses the results of the experiments, which were conducted in a Python 3.8 environment on Ubuntu 20.04, utilizing PyTorch 1.10 along with the sklearn 1.1.3, Keras, and matplotlib libraries.

## Implementation details

To demonstrate the superior performance of the ensemble learning model developed in this study for predicting learning achievement, we compare it with six independent prediction models. This section provides a detailed description of the machine learning models used, including their configurations and training processes.

- KNN: We set the number of neighbors (k) to 5 and used Euclidean distance as the metric for calculating distances between points.

- NB: We used the Gaussian Naive Bayes variant, which is suitable for continuous data.

- RF: We employed 100 trees with the Gini impurity as the splitting criterion.

- GBDT: We used a learning rate of 0.1 and 100 boosting stages.

- XGBoost: The learning rate was set to 0.1, with a maximum depth of 4 and 100 boosting rounds.

- MLP: We configured the MLP with one hidden layer consisting of 100 neurons, ReLU activation functions, and used the Adam optimizer for training.

- LR: We employed L2 regularization to prevent overfitting.

The detailed parameter settings and values for each model are summarized in Table 3.

In machine learning, various model evaluation strategies, such as k-fold cross-validation (CV), jackknife, and independent testing, are employed to assess the performance of prediction models. However, the jackknife test is often constrained by its extensive computational time and the large number of calculations required. To address these limitations and improve the model's generalization capability while avoiding overfitting, this study utilizes the k-fold CV method. Specifically, the training dataset is randomly divided into k non-overlapping, approximately equal-sized subsets. The model is trained on k-1 subsets and tested on the remaining subset in each iteration. For this study, k was set to 5. The dataset was initially split into training and test sets with an 8:2 ratio, and 5-fold cross-validation was used to tune hyperparameters and validate the models.

## Experimental evaluation metric

In the domain of classification tasks, evaluating the effectiveness of a model requires employing robust evaluation metrics. Precision, Recall, F1, and Accuracy are among the most

**Table 3. Related parameter settings of each model.**

| Model | Parameter Settings and parameter values |
|---|---|
| KNN | n_neighbors = 5,weights='uniform',algorithm='auto',metric='euclidean' |
| NB | var_smoothing = 1e-09 |
| RF | n_estimators = 100,criterion='gini',max_depth = 5 |
| GBDT | learning_rate = 0.1,n_estimators = 100,min_samples_split = 2,min_samples_leaf = 1, |
| XGboost | learning_rate = 0.1,n_estimators = 100,subsample = 1.0,max_depth = 4 |
| MLP | hidden_layer_sizes=(100,), activation='relu', solver='adam',learning_rate = 0.001 |
| LR | penalty='l2',C = 1.0, solver='lbfgs',max_iter = 100 |

commonly used quantitative measures to assess a classifier's performance (see Eqs (3), (4), (5) and (6).

$$\text{Precision} = \frac{TP}{TP + FP} \tag{3}$$

$$\text{Recall} = \frac{TP}{TP + FN} \tag{4}$$

$$F1 = \frac{2 \times \text{Recall} \times \text{Precision}}{\text{Recall} + \text{Precision}} \tag{5}$$

$$\text{Accuracy} = \frac{TP + TN}{TP + TN + FP + FN} \tag{6}$$

## Experimental results

**Model performance evaluation.** This study conducts a comparative analysis of a learning achievement prediction method based on stacking ensemble learning compare with six independent machine learning models, underscoring the advantages of the stacking ensemble approach. The six base learning models utilized are KNN, NB, RF, GBDT, XGBoost, and MLP, with Logistic Regression (LR) serving as the meta model to form a Stacking classifier. To address the challenges posed by imbalanced datasets during the experiments, various sampling techniques were employed.

Detailed results and insights derived from the classifier performance are presented in Table 4. Overall, the models demonstrated satisfactory accuracy in predicting student learning achievements, with most achieving approximately 0.9 on the test set. Notably, the stacking ensemble learning models outperformed the six independent machine learning models. Specifically, the ensemble learning model utilizing the OneSidedSelection resampling strategy achieved an accuracy of 0.8520 (compared to NB: 0.6751, KNN: 0.7882, RF: 0.8395, GBDT: 0.8282, XGBoost: 0.8265, and MLP: 0.8048), surpassing all other independent models. This model also demonstrated superior performance metrics, with a higher F1 (0.8597) and precision (0.9853) compared to the independent models.

Receiver Operating Characteristic (ROC) curves are essential tools for assessing the predictive performance of models, particularly in the context of predicting learning achievement. As shown in Fig 6, our proposed ensemble learning model achieves an impressive Area Under the Curve (AUC) of 0.9953, outperforming both the XGBoost and MLP models. The ROC curve's diagonal line serves as a reference, marking the distinction between true positives and false negatives [50]. This analysis highlights the superior accuracy and performance of stacking ensemble learning models in educational contexts and demonstrates the effectiveness of the stacking ensemble approach compared to individual machine learning models. The findings provide valuable insights into the benefits of ensemble techniques for predicting learning outcomes.

*Baseline approaches.* In addition to comparing against the individual machine learning models that make up the Stacking ensemble (i.e., KNN, NB, RF, GBDT, XGBoost, and MLP), we also conducted comparative experiments with state-of-the-art models (i.e., Song et al. [51], Liu et al. [52], Zerkouk et al. [53]). Below is a brief overview of the baselines:

**Table 4. Learning achievement prediction results.**

| Resampling type | Resampling method | Models | Precision | Recall | F1 | Accuracy |
|---|---|---|---|---|---|---|
| Oversampling | ROS | NB | 0.6596 | 0.9191 | 0.7680 | 0.9711 |
| | | KNN | 0.6905 | 0.9393 | 0.7960 | 0.9749 |
| | | RF | 0.6524 | 0.9835 | 0.7845 | 0.9719 |
| | | GBDT | 0.6725 | 0.9816 | 0.7982 | 0.9742 |
| | | XGBoost | 0.7496 | 0.9412 | 0.8346 | 0.9806 |
| | | MLP | 0.7179 | 0.9449 | 0.8159 | 0.9778 |
| | | Stacking | 0.8417 | 0.7114 | 0.7914 | 0.9805 |
| | SMOTE | NB | 0.6636 | 0.9320 | 0.7752 | 0.9719 |
| | | KNN | 0.6745 | 0.9485 | 0.7884 | 0.9735 |
| | | RF | 0.6782 | 0.9724 | 0.7991 | 0.9745 |
| | | GBDT | 0.7052 | 0.9761 | 0.8188 | 0.9775 |
| | | XGBoost | 0.8063 | 0.8952 | 0.8484 | 0.9833 |
| | | MLP | 0.7184 | 0.9191 | 0.8065 | 0.9770 |
| | | Stacking | 0.8284 | 0.8787 | 0.8528 | 0.9842 |
| | ADASYN | NB | 0.6278 | 0.9301 | 0.7496 | 0.9676 |
| | | KNN | 0.6520 | 0.9504 | 0.7734 | 0.9710 |
| | | RF | 0.5882 | 0.9926 | 0.7387 | 0.9634 |
| | | GBDT | 0.6304 | 0.9908 | 0.7706 | 0.9693 |
| | | XGBoost | 0.7744 | 0.8897 | 0.8281 | 0.9808 |
| | | MLP | 0.7010 | 0.9265 | 0.7981 | 0.9756 |
| | | Stacking | 0.7887 | 0.8989 | 0.8402 | 0.9822 |
| Undersampling | RUS | NB | 0.6179 | 0.9007 | 0.7330 | 0.9658 |
| | | KNN | 0.6118 | 0.9761 | 0.7521 | 0.9665 |
| | | RF | 0.6456 | 0.9779 | 0.7778 | 0.9709 |
| | | GBDT | 0.6601 | 0.9890 | 0.7918 | 0.9729 |
| | | XGBoost | 0.6520 | 0.9779 | 0.7824 | 0.9717 |
| | | MLP | 0.6487 | 0.9743 | 0.7788 | 0.9712 |
| | | Stacking | 0.6604 | 0.9761 | 0.7878 | 0.9726 |
| | Tomek-Links | NB | 0.6649 | 0.9007 | 0.7650 | 0.9712 |
| | | KNN | 0.7882 | 0.8621 | 0.8235 | 0.9808 |
| | | RF | 0.8402 | 0.8603 | 0.8501 | 0.9842 |
| | | GBDT | 0.8284 | 0.8787 | 0.8528 | 0.9842 |
| | | XGBoost | 0.8316 | 0.8897 | 0.8597 | 0.9849 |
| | | MLP | 0.8120 | 0.8971 | 0.8524 | 0.9838 |
| | | Stacking | 0.8445 | 0.8787 | 0.8513 | 0.9843 |
| | EditedNearestNeighbours | NB | 0.5797 | 0.9430 | 0.7018 | 0.9614 |
| | | KNN | 0.7538 | 0.9118 | 0.8253 | 0.9799 |
| | | RF | 0.7771 | 0.9099 | 0.8383 | 0.9817 |
| | | GBDT | 0.7635 | 0.9375 | 0.8416 | 0.9816 |
| | | XGBoost | 0.7658 | 0.9375 | 0.8430 | 0.9818 |
| | | MLP | 0.7533 | 0.9375 | 0.8354 | 0.9808 |
| | | Stacking | 0.7764 | 0.9320 | 0.8471 | 0.9825 |
| | OneSidedSelection | NB | 0.6751 | 0.8824 | 0.7649 | 0.9718 |
| | | KNN | 0.7882 | 0.8621 | 0.8235 | 0.9808 |
| | | RF | 0.8395 | 0.8750 | 0.8569 | 0.9848 |
| | | GBDT | 0.8282 | 0.8860 | 0.8561 | 0.9845 |
| | | XGBoost | 0.8265 | 0.8934 | 0.8587 | 0.9847 |
| | | MLP | 0.8048 | 0.9173 | 0.8574 | 0.9841 |
| | | Stacking | 0.8520 | 0.8676 | 0.8597 | 0.9853 |

(*Continued*)

**Table 4.** (Continued)

| Resampling type | Resampling method | Models | Precision | Recall | F1 | Accuracy |
|---|---|---|---|---|---|---|
| Hybrid Sampling | SMOTEENN | NB | 0.5657 | 0.9577 | 0.7113 | 0.9595 |
| | | KNN | 0.6489 | 0.9614 | 0.7748 | 0.9709 |
| | | RF | 0.6533 | 0.9871 | 0.7862 | 0.9720 |
| | | GBDT | 0.6785 | 0.9816 | 0.8024 | 0.9748 |
| | | XGBoost | 0.7305 | 0.9467 | 0.8247 | 0.9790 |
| | | MLP | 0.7030 | 0.9485 | 0.8075 | 0.9765 |
| | | Stacking | 0.7067 | 0.9522 | 0.8113 | 0.9769 |
| | SMOTETomek | NB | 0.6619 | 0.9320 | 0.7740 | 0.9717 |
| | | KNN | 0.6675 | 0.9485 | 0.7836 | 0.9727 |
| | | RF | 0.6658 | 0.9816 | 0.7935 | 0.9734 |
| | | GBDT | 0.6996 | 0.9761 | 0.8150 | 0.9769 |
| | | XGBoost | 0.8010 | 0.8952 | 0.8455 | 0.9830 |
| | | MLP | 0.7290 | 0.9393 | 0.8209 | 0.9787 |
| | | Stacking | 0.8247 | 0.8732 | 0.8482 | 0.9837 |

- Song et al. [51]: This study employs a variant of the Grey Wolf Optimization (GWO) algorithm to optimize the weights and biases of Multi-Layer Perceptron (MLP) models for predicting student achievement.

- Liu et al. [52]: This approach integrates Bi-LSTM with attention mechanisms and LightGBM to predict MOOCs dropouts by effectively modeling both time series and general information features.

- Zerkouk et al. [53]: This model uses XGBoost in combination with logistic regression to develop a binary classification framework that accurately predicts student dropout by analyzing socio-demographic and behavioral data.

The results, as summarized in Table 5 using the XuetangX dataset, consistently demonstrate that our proposed model outperforms the other models across key performance metrics, achieving the highest precision (0.8520), recall (0.8676), F1 (0.8597), and accuracy (0.9853). While the other models show effectiveness in specific areas, they each have limitations that impact their overall performance. For instance, the model in [52] does not adequately address data imbalance, primarily focusing on video features while overlooking critical factors such as student profiles. Similarly, the model in [53] relies exclusively on the XGBoost algorithm, limiting its adaptability by not leveraging the potential benefits of ensemble methods. Additionally, [51] confines its approach to using an MLP model for predicting student performance, thereby missing the advantages of integrating multiple algorithms. Furthermore, none of these models sufficiently tackle the crucial issue of model interpretability, which is vital for enhancing educational outcomes and aiding informed decision-making.

Our proposed stacking ensemble model effectively overcomes these limitations by integrating the strengths of various models. By employing a meta learner, such as Logistic Regression (LR), to aggregate the outputs of multiple base learners, the model successfully balances the simplicity of linear models with the complexity of non-linear ones, resulting in superior overall performance. Additionally, our approach directly addresses the issue of data imbalance, ensuring more accurate and reliable predictions. We also emphasize model interpretability, which is crucial for deriving actionable insights in educational settings. This strategic integration

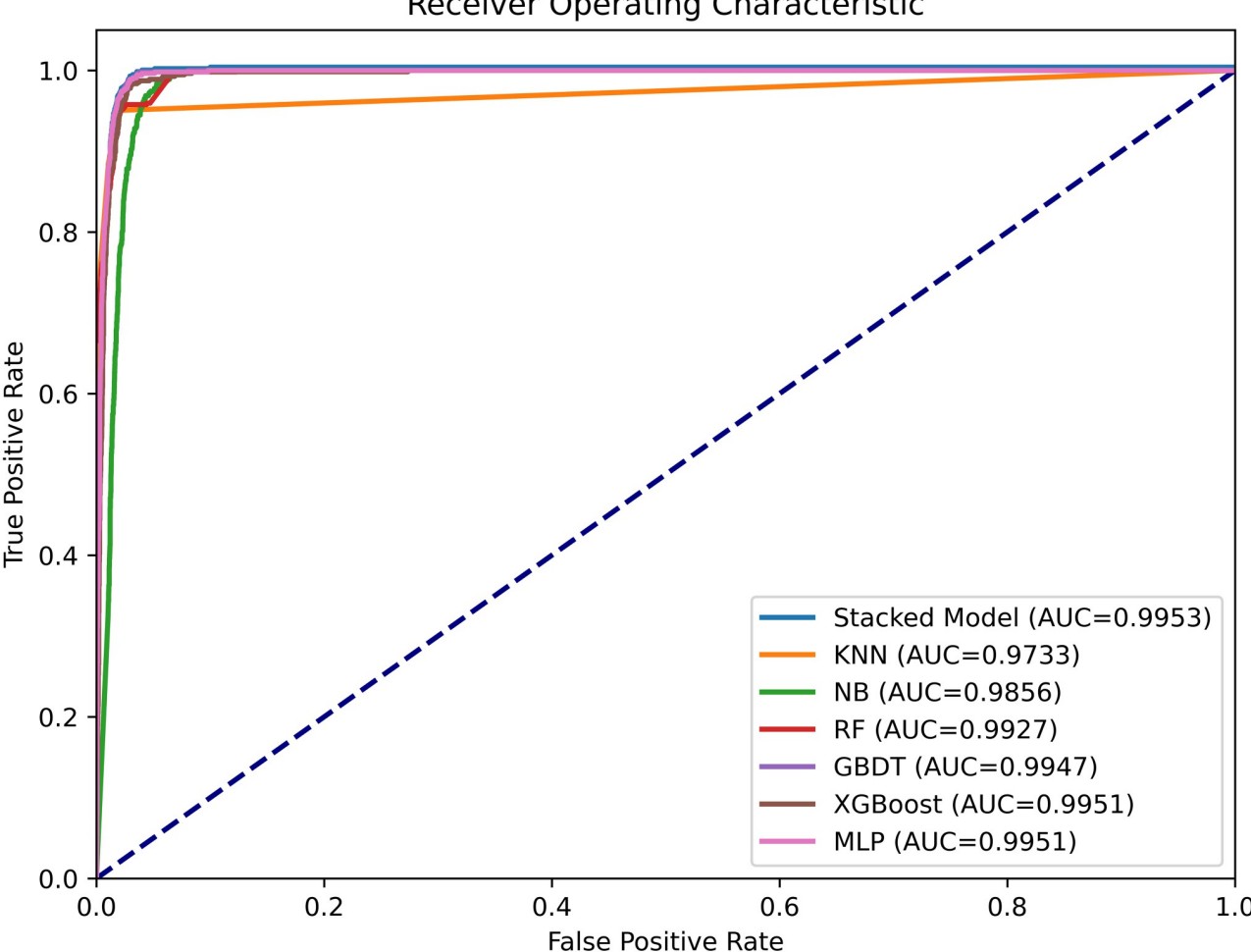

**Fig 6. Receiver operating characteristic curve.**

enables our method to outperform individual models, providing more dependable and precise predictions of learning outcomes.

**Model explanation based on SHAP.** This section presents the results of the interpretability analysis conducted using the SHAP method. To illustrate the specific influence of each feature on the prediction of learning achievement, a SHAP summary plot for each feature is introduced (Fig 7).

In Fig 7, the vertical axis represents the Shapley value for each feature, while the horizontal axis shows the distribution of these values across the samples. Each point on the graph

**Table 5. Comparison of results with the existing model.**

| Models | Precision | Recall | F1 | Accuracy |
|---|---|---|---|---|
| Song et al. [51] | 0.8347 | 0.8262 | 0.8304 | 0.9634 |
| Liu et al. [52] | 0.8285 | 0.8104 | 0.8249 | 0.9664 |
| Zerkouk et al. [53] | 0.8396 | 0.8294 | 0.8316 | 0.9703 |
| Proposed Model | 0.8520 | 0.8676 | 0.8597 | 0.9853 |

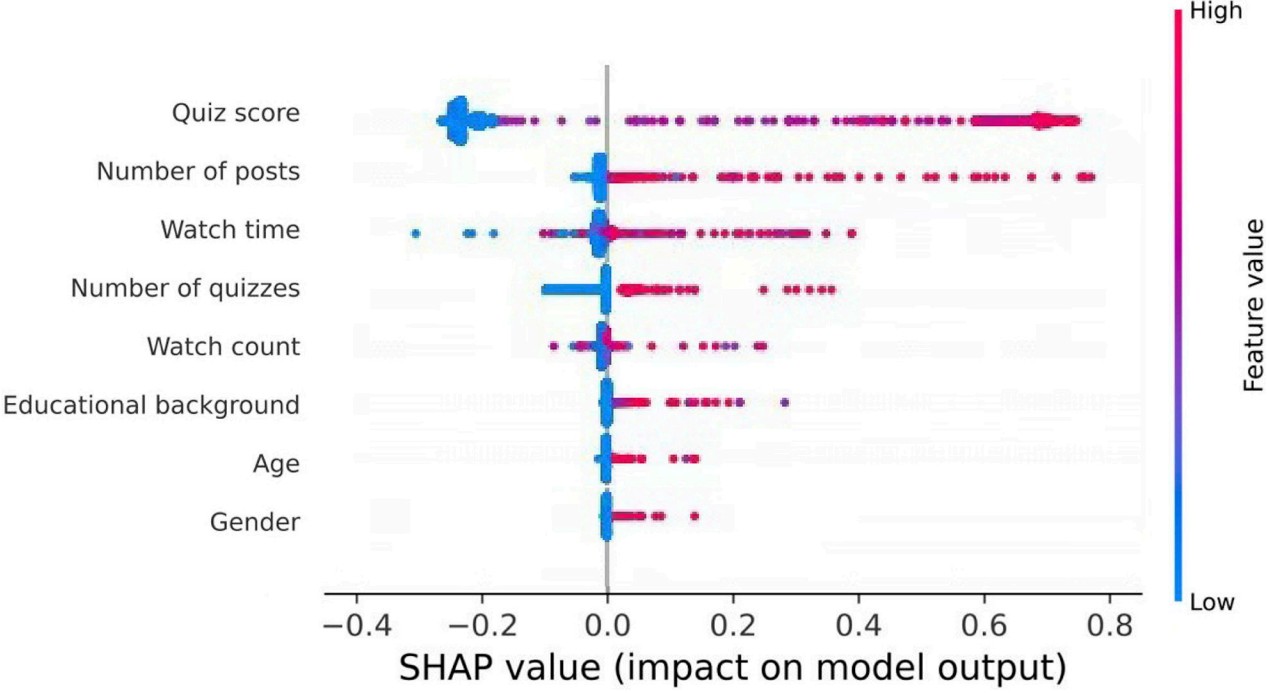

**Fig 7. Model explanation through SHAP.**

corresponds to a sample, with the color gradient (ranging from blue to red) indicating the feature's value. The middle axis at a SHAP value of 0 signifies minimal feature importance, where points tend to cluster. The vertical order of the features reflects their relative importance, with the most influential features listed at the top, descending in significance toward the bottom.

The SHAP summary graph for each feature serves two primary purposes: demonstrating global feature importance and illustrating how feature values influence predictions of learning achievement. As shown in Fig 7, quiz score, number of posts, watch time, number of quizzes, and watch counts emerge as the top five influential factors in predicting learning achievement. For instance, a higher quiz score is strongly associated with a greater likelihood of course completion. Similarly, the number of quizzes, as indicated by the number of completed tests, shows a positive correlation with learning performance [25, 54, 55]. The number of posts ranks second, with Fig 7 confirming that increased posting activity enhances completion probabilities, aligning with previous studies [56, 57]. Additionally, watch time and watch counts, which reflect the duration and frequency of video viewing, demonstrate that greater engagement with video content is positively correlated with student success in courses. These findings highlight the critical role of active participation and focused engagement in shaping learning outcomes, as revealed through SHAP analysis.

To evaluate the importance of SHAP-identified high-rank features on model performance, we conducted a comprehensive ablation study. This involved systematically removing each high-rank feature, retraining the model, and analyzing the impact on key performance metrics, including precision, recall, F1, and accuracy. The findings, as shown in Table 6, highlight the crucial role these features play in maintaining the model's predictive accuracy and generalization capability.

The results clearly demonstrate that SHAP-based high-rank features are vital for the model's performance. The removal of any top-ranked feature leads to a noticeable decline in all

**Table 6. Experimental results after removing high rank features.**

| Feature removal | Precision | Recall | F1 | Accuracy |
|---|---|---|---|---|
| Quiz score | 0.8207 | 0.8106 | 0.8154 | 0.9609 |
| Number of posts | 0.8231 | 0.8025 | 0.8161 | 0.9648 |
| Watch time | 0.8336 | 0.8474 | 0.8405 | 0.9732 |
| Number of quizzes | 0.8371 | 0.8695 | 0.8541 | 0.9741 |
| Watch count | 0.8403 | 0.8671 | 0.8553 | 0.9747 |
| Educational background | 0.8392 | 0.8676 | 0.8547 | 0.9753 |
| Age | 0.8479 | 0.8713 | 0.8595 | 0.9802 |
| Gender | 0.8481 | 0.8621 | 0.8551 | 0.9818 |

metrics, underscoring their indispensability. For instance, the elimination of critical features like quiz score and the number of posts results in significant performance drops, emphasizing their importance in sustaining predictive accuracy. On the other hand, removing features like age and gender had minimal or even positive effects, suggesting that these factors might contribute to unnecessary model complexity.

Overall, the study confirms that SHAP-identified high-rank features are essential for ensuring the model's robustness and high predictive accuracy. Their retention is necessary for achieving effective generalization in educational data mining models.

Beyond the technical findings, these insights have practical applications for educators. By focusing on the top-ranked features, educators can design more targeted instructional materials, develop personalized learning plans, and optimize learning activities. For example, if student engagement with interactive content is identified as a key factor, incorporating more quizzes and simulations into lessons could enhance learning outcomes. Additionally, SHAP analysis can inform feedback and assessment strategies, ensuring that feedback is both timely and aligned with the most impactful factors on student success.

The insights from this study can also guide professional development for educators, helping them incorporate these findings into their teaching practices. For instance, workshops on creating engaging content or managing discussion forums can leverage these insights to improve educational practices.

In summary, integrating SHAP-based feature analysis into educational settings not only enhances the practical relevance of our study but also demonstrates its broader impact, enabling educators to create more effective, personalized, and engaging learning environments tailored to their students' needs.

## Conclusions and limitations

Numerous studies have utilized early prediction methodologies to predict student performance through machine learning and statistical analyses [58–62]. However, these efforts have primarily concentrated on identifying the most influential features for predicting student learning achievement. In contrast, our proposed method not only achieves high accuracy in predicting learning performance but also provides interpretable machine learning outputs, offering valuable insights into the factors influencing student achievement, even for non-experts.

This research addresses key limitations of previous studies, such as inaccuracies in dropout prediction and the lack of interpretability in prediction results, by introducing a novel approach using ensemble learning. Specifically, our stacked ensemble learning technique

integrates data from students' online learning behavior logs and demographic information, resulting in a predictive model with an impressive accuracy of 98.53%. Through SHAP value analysis, we examined the impact of various features on student dropout rates, revealing that interactions within learning activities—such as video resource usage, quiz participation, and forum engagement—significantly influence dropout rates more than demographic factors.

While the proposed algorithm demonstrates robust and accurate predictive outcomes, especially with a large number of predictors, it is important to acknowledge its limitations. The current research primarily relies on static data sources, lacking comprehensive multimodal data collection and analysis. This limitation hinders the capture of implicit higher-order features, such as learners' motivation, cognitive engagement, and learning styles, which are dynamic and context-dependent. To fully capture these features, specialized models like recurrent neural networks (RNNs) or temporal convolutional networks (TCNs) are needed, as they are designed to handle time-varying data.

Moreover, while the model shows high predictive accuracy within the scope of this study, its generalizability to different educational contexts and diverse student populations is yet to be established. Further validation with varied datasets and educational settings is necessary to ensure the model's robustness and applicability across different scenarios.

In future research, we plan to improve prediction performance by developing more sophisticated features using deep learning models. We will incorporate established theories such as inquiry community theory, self-determination theory, and the technology acceptance model to gather comprehensive multimodal datasets from online learning environments. These datasets will enable us to apply deep learning techniques to extract nuanced features like learner motivation and learning style, enhancing our ability to identify and predict student dropout risks early. Additionally, we intend to implement this framework as an automated solution for academic institutions, validating its effectiveness in real-world educational settings. For instance, the system could proactively alert at-risk students and provide educators with actionable recommendations for timely interventions.

## Acknowledgments

The authors sincerely thank the anonymous reviewers and editors for their valuable feedback. We also extend our gratitude to the organizers of XuetangX for providing access to the datasets used in this study.

## Author Contributions

**Conceptualization:** Tingting Tong.

**Data curation:** Zhen Li.

**Formal analysis:** Zhen Li.

**Funding acquisition:** Zhen Li.

**Investigation:** Zhen Li.

**Methodology:** Tingting Tong.

**Software:** Tingting Tong.

**Validation:** Zhen Li.

**Visualization:** Tingting Tong.

**Writing – original draft:** Tingting Tong.

**Writing – review & editing:** Tingting Tong, Zhen Li.

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
