## [Decision Letter · Decision Letter 0]

29 Jul 2024

PONE-D-24-26784Predicting Learning Achievement Using Ensemble Learning with Result ExplanationPLOS ONE

Dear Dr. Tong,

Thank you for submitting your manuscript to PLOS ONE. After careful consideration, we feel that it has merit but does not fully meet PLOS ONE’s publication criteria as it currently stands. Therefore, we invite you to submit a revised version of the manuscript that addresses the points raised during the review process.

We look forward to receiving your revised manuscript.

Kind regards,

Shahid Akbar, PhD

Academic Editor

PLOS ONE

Journal Requirements:

Additional Editor Comments: 

**Major Revision**

Reviewers' comments:

Reviewer's Responses to Questions

**Comments to the Author**

1. Is the manuscript technically sound, and do the data support the conclusions?

Reviewer #1: Partly

Reviewer #2: Yes

2. Has the statistical analysis been performed appropriately and rigorously? 

Reviewer #1: Yes

Reviewer #2: Yes

3. Have the authors made all data underlying the findings in their manuscript fully available?

Reviewer #1: Yes

Reviewer #2: Yes

4. Is the manuscript presented in an intelligible fashion and written in standard English?

Reviewer #1: Yes

Reviewer #2: Yes

5. Review Comments to the Author

Reviewer #1: 1. The authors should mention the main contributions and novelty at the end of introduction section.

2. The authors mentioned the stacking ensemble based learning model but it's not properly explained I suggest providing a section related stack ensemble learning with its properl procedure.

3. For the reader concerns the authors should incorporate the recent predictors such as DeepAVP-TPPred, Deepstacked-AVPs, iAFPs-Mv-BiTCN, cACP, iACP-GAEnsC, and pAVP_PSSMDWT-EnC.

4. How the authors handle overfitting of the proposed training model.

5. I didn't found comparison of the proposed model with existing state of the art methods to validateits effectiveness.

6. The authors used the SHAP based feature analysis to evaluate feature contributions. However, it would be more conceiving if the authors discuss the Shap based high rank features and its effectiveness to model performance and real life implementation.

Reviewer #2: 1. How the generalization of the proposed model was ensured in terms of model overfitting.

2. The authors should provide the pseudocode (algorithm) of the proposed model.

3. The applied machine learning models are needs to explained in an extra section like training models.

4. Expanding the comparison with existing methods is crucial. Strengthening this comparison will help accentuate the unique advantages offered by the proposed model.

5. What are the limitations of the proposed model.

6. PLOS authors have the option to publish the peer review history of their article (what does this mean?). If published, this will include your full peer review and any attached files.

Reviewer #1: No

Reviewer #2: No

---

## [Author Response · Author response to Decision Letter 0]

2 Sep 2024

Original Manuscript ID: PONE-D-24-26784

Original Article Title: “Predicting Learning Achievement Using Ensemble Learning with Result Explanation” 

Dear Editor,

Thank you for allowing a resubmission of our manuscript, with an opportunity to address the reviewers’ comments.

We are uploading (a) our point-by-point response to the comments (below) (response to reviewers), (b) an updated manuscript with yellow highlighting indicating changes. and (c) a clean updated manuscript without highlights (PDF main document).

We sincerely hope this manuscript will be accepted for publication on PLOS ONE.

Best regards,

Tingting Tong

Reviewer#1, Concern # 1: The authors should mention the main contributions and novelty at the end of the introduction section.

Author response: We appreciate your feedback on the importance of clearly highlighting the main contributions and novelty of our work in the introduction section.

In response to your suggestion, we have revised the introduction to clearly state the primary contributions and novel aspects of our research. Specifically, we have added a new paragraph at the end of the introduction, located between lines 36 and 52 in the revised manuscript.

This paragraph concisely outlines the main contributions of our research, emphasizing their significance and novelty in the field. We believe that this addition enhances the clarity and impact of our introduction. Thank you once again for your valuable insights.

Reviewer#1, Concern # 2: The authors mentioned the stacking ensemble-based learning model but it's not properly explained. I suggest providing a section related to stack ensemble learning with its proper procedure.

Author response: We sincerely apologize for the unclear expression in the original manuscript. We appreciate your insightful observation regarding the need for a more comprehensive explanation of the stacking ensemble-based learning model. We recognize that providing a detailed description of this model and its procedure is essential for readers to fully understand the methodology and its implications.

In response to your suggestion, we have added a new section specifically dedicated to explaining the stacking ensemble learning model, located between lines 314 and 330 in the revised manuscript.

We hope this addresses your concerns effectively. Thank you once again for your thorough review and constructive comments.

Reviewer#1, Concern # 3: For the reader concerns the authors should incorporate the recent predictors such as DeepAVP-TPPred, Deepstacked-AVPs, iAFPs-Mv-BiTCN, cACP, iACP-GAEnsC, and pAVP_PSSMDWT-EnC.

Author response: Thank you for highlighting the importance of incorporating recent predictors in our analysis. We appreciate your suggestion to incorporate recent models like DeepAVP-TPPred, Deepstacked-AVPs, iAFPs-Mv-BiTCN, cACP, iACP-GAEnsC, and pAVP_PSSMDWT-EnC, which represent significant advancements in predictive modeling.

Recognizing the importance of these contributions, we have expanded our related work section to include a thorough discussion of these advanced predictors. The revised content is located between lines 129 and 140.

Thank you once again for your constructive suggestions, which have undoubtedly improved the level of our manuscript. 

Reviewer#1, Concern # 4: How the authors handle overfitting of the proposed training model.

Author response: We appreciate your concern regarding how overfitting is managed in our proposed training model. To mitigate overfitting, we implemented two key strategies.

First, we utilized a simple linear regression model as the meta-learner, which helps to mitigate model overfitting. The details are located between lines 303 and 309.

Second, we employed k-fold cross-validation to address the problem of model overfitting.

By using k-fold cross-validation, the model does not rely solely on a specific training set. Instead, it is trained and evaluated multiple times on different training sets, thereby reducing its dependence on any particular training data. This repeated training process helps the model learn more generalizable features, preventing it from overfitting to a specific training set. The specifics are located between lines 476 and 486.

By incorporating these strategies, we aimed to effectively reduces the risk of overfitting, making the model’s training process more robust and ensuring stronger generalization capabilities.

Thank you once again for your thorough review and constructive comments.

Reviewer#1, Concern # 5: I didn't find a comparison of the proposed model with existing state-of-the-art methods to validate its effectiveness.

Author response: Thank you for your insightful advice. In response to your feedback, we have added three state-of-the-art baselines to our analysis. These baselines represent the most recent and advanced methods in the field, providing a robust benchmark for evaluating the effectiveness of our proposed model. The comparison results are located between lines 521 and 557.

We hope this addresses your concerns effectively. Thank you again for your feedback, which is invaluable in improving the quality of our work.

Reviewer#1, Concern # 6: The authors used the SHAP-based feature analysis to evaluate feature contributions. However, it would be more convincing if the authors discuss the SHAP-based high-rank features and their effectiveness in model performance and real-life implementation.

Author response: Thank you for your insightful comments and suggestions. We appreciate your feedback on our manuscript and are pleased to address your concerns regarding the SHAP-based feature analysis.

You raised a valid point about discussing the SHAP-based high-ranking features and their effectiveness in model performance and real-life implementation. We agree that this would enhance the clarity and applicability of our findings.

In response, we conducted ablation experiment to understand the significance of SHAP-based high-rank features on model performance. Specifically, we sequentially removed each of the top-ranked features identified by SHAP analysis, retrained the model, and recorded the performance metrics. The results are located between lines 583 and 614 in the revised manuscript.

These findings provide strong evidence that the SHAP-based high-rank features identified in our analysis are indeed crucial for the model's performance. Moreover, they highlight the potential for real-world applications, as these features can be used to prioritize data collection and feature engineering efforts. Thank you once again for your valuable insights.

Original Manuscript ID: PONE-D-24-26784

Original Article Title: “Predicting Learning Achievement Using Ensemble Learning with Result Explanation” 

Dear Editor,

Thank you for allowing a resubmission of our manuscript, with an opportunity to address the reviewers’ comments.

We are uploading (a) our point-by-point response to the comments (below) (response to reviewers), (b) an updated manuscript with yellow highlighting indicating changes. and (c) a clean updated manuscript without highlights (PDF main document).

We sincerely hope this manuscript will be accepted for publication on PLOS ONE.

Best regards,

Tingting Tong

Reviewer#2, Concern # 1: How the generalization of the proposed model was ensured in terms of model overfitting.

Author response: Thank you for your valuable feedback regarding the generalization of the proposed model. We understand the importance of addressing model overfitting to ensure robust performance. In our study, we employed two techniques to mitigate overfitting, which is shown as follows,

First, we employed a simple linear regression model as the meta-learner, which helps to avoid overfitting in the final prediction. The details are located between lines 303 and 309 in the revised manuscript.

Second, we used k-fold cross-validation to address the problem of model overfitting.

By using k-fold cross-validation, the model does not rely solely on a specific training set. Instead, it is trained and evaluated multiple times on different training sets, thereby reducing its dependence on any particular training data. This repeated training process helps the model learn more generalizable features, preventing it from overfitting to a specific training set. The specifics are detailed below. The details are located between lines 476 and 486 in the revised manuscript.

We hope this addresses your concerns effectively. Thank you again for your feedback, which is invaluable in improving the quality of our work.

Reviewer#2, Concern # 2: The authors should provide the pseudocode (algorithm) of the proposed model.

Author response: Thank you for your insightful advice. Providing pseudocode will indeed help readers understand the implementation details and reproduce our results. 

In response, we have included the pseudocode for the proposed model in the revised manuscript, located between lines 385 and 416.

Thank you once again for your constructive suggestions, which have undoubtedly improved the quality of our manuscript. 

Reviewer#2, Concern # 3: The applied machine learning models need to be explained in an extra section like training models.

Author response: Thank you for your valuable feedback and for highlighting the need to provide a more detailed explanation of the applied machine learning models. We fully agree that a thorough understanding of the models and their training processes is crucial for the readers.

In response to your suggestion, we have included a thorough explanation of the machine learning models applied in our study in the revised manuscript. This enhancement offers a detailed account of each model used, covering aspects such as the training processes, hyperparameter settings, and model configurations. These details can be found between lines 354-384 and 454-486.

We believe that this addition will offer greater clarity and insight into the methodology of our research, addressing the concern you raised.

Thank you once again for your constructive suggestions, which have undoubtedly improved the quality of our manuscript.

Reviewer#2, Concern # 4: Expanding the comparison with existing methods is crucial. Strengthening this comparison will help accentuate the unique advantages offered by the proposed model.

Author response: Thank you for your insightful advice. We appreciate your valuable feedback and believe that these additions will clearly demonstrates the unique advantages of our proposed model over existing methods. In response to your advice, we expand the comparison section in our revised manuscript, located between lines 521 and 557.

Thank you once again for your valuable insights.

Reviewer#2, Concern # 5: What are the limitations of the proposed model.

Author response: Thank you for highlighting the importance of discussing the limitations of our proposed model. We acknowledge that providing a balanced view that includes the limitations of our research is crucial for transparency and scientific rigor. 

In response to your feedback, we have added a detailed discussion of the limitations in the revised manuscript. These limitations are specifically addressed between lines 632 and 645.

We believe that by openly discussing these limitations, we provide a more comprehensive and balanced perspective on our research. Moreover, we highlight that these limitations offer valuable opportunities for future research and development, where our work can be further refined and enhanced. We appreciate your insightful feedback once again.

---

## [Decision Letter · Decision Letter 1]

8 Sep 2024

PONE-D-24-26784R1Predicting Learning Achievement Using Ensemble Learning with Result ExplanationPLOS ONE

Dear Dr. Tong,

Thank you for submitting your manuscript to PLOS ONE. After careful consideration, we feel that it has merit but does not fully meet PLOS ONE’s publication criteria as it currently stands. Therefore, we invite you to submit a revised version of the manuscript that addresses the points raised during the review process.

**ACADEMIC EDITOR: Major Revision **

We look forward to receiving your revised manuscript.

Kind regards,

Shahid Akbar, PhD

Academic Editor

PLOS ONE

Reviewers' comments:

Reviewer's Responses to Questions

**Comments to the Author**

1. If the authors have adequately addressed your comments raised in a previous round of review and you feel that this manuscript is now acceptable for publication, you may indicate that here to bypass the “Comments to the Author” section, enter your conflict of interest statement in the “Confidential to Editor” section, and submit your "Accept" recommendation.

Reviewer #1: (No Response)

Reviewer #2: All comments have been addressed

2. Is the manuscript technically sound, and do the data support the conclusions?

Reviewer #1: Yes

Reviewer #2: Yes

3. Has the statistical analysis been performed appropriately and rigorously? 

Reviewer #1: Yes

Reviewer #2: Yes

4. Have the authors made all data underlying the findings in their manuscript fully available?

Reviewer #1: Yes

Reviewer #2: Yes

5. Is the manuscript presented in an intelligible fashion and written in standard English?

Reviewer #1: Yes

Reviewer #2: Yes

6. Review Comments to the Author

Reviewer #1: In the revised version, the references (32-37) are incorrectly cited. I suggest authors to carefully address such issue according to the following doi links:

Deepstacked-AVPs
doi.org/10.1186/s12859-024-05726-5

iAFPs-Mv-BiTCN
doi.org/10.1016/j.artmed.2024.102860

Prediction of Antiviral Peptides Using Transform Evolutionary & SHAP Analysis Based Descriptors by Incorporation with Ensemble Learning Strategy. �
doi.org/10.1016/j.chemolab.2022.104682

iACP-GAEnsC
doi.org/10.1016/j.artmed.2017.06.008

CACP
doi.org/10.1016/j.chemolab.2019.103912

DeepAVP-TPPred �
doi.org/10.1093/bioinformatics/btae305

Reviewer #2: my required comments are successfully incorporated by the authors and now the quality of the paper have significantly been improved.

7. PLOS authors have the option to publish the peer review history of their article (what does this mean?). If published, this will include your full peer review and any attached files.

Reviewer #1: No

Reviewer #2: No

---

## [Author Response · Author response to Decision Letter 1]

19 Sep 2024

Manuscript Revision ID: PONE-D-24-26784R1

Original Article Title: “Predicting Learning Achievement Using Ensemble Learning with Result Explanation” 

Dear Editor,

Thank you for allowing a resubmission of our manuscript, with an opportunity to address the reviewers’ comments.

We are uploading (a) our point-by-point response to the comments (below) (response to reviewers), (b) an updated manuscript with yellow highlighting indicating changes. and (c) a clean updated manuscript without highlights (PDF main document).

We sincerely hope this manuscript will be accepted for publication on PLOS ONE.

Best regards,

Tingting, Tong

Reviewer#1, Concern # 1: In the revised version, the references (32-37) are incorrectly cited. I suggest authors to carefully address such issue according to the following doi links:

Deepstacked-AVPs: doi.org/10.1186/s12859-024-05726-5

iAFPs-Mv-BiTCN: doi.org/10.1016/j.artmed.2024.102860

Prediction of Antiviral Peptides Using Transform Evolutionary & SHAP Analysis Based Descriptors by Incorporation with Ensemble Learning Strategy: doi.org/10.1016/j.chemolab.2022.104682

iACP-GAEnsC: doi.org/10.1016/j.artmed.2017.06.008

CACP: doi.org/10.1016/j.chemolab.2019.103912

DeepAVP-TPPred: doi.org/10.1093/bioinformatics/btae305

Author response: We greatly appreciate the time and effort you have invested in reviewing our manuscript, and we are thankful for your valuable comments and suggestions.

We sincerely apologize for the incorrect DOIs cited in the revised version of our manuscript. After carefully reviewing the references, we have corrected the DOIs in line with your recommendations. The references now accurately reflect the correct citations, with the updated DOI links as follows: 

32. Akbar S, Raza A, Zou Q. Deepstacked-AVPs: predicting antiviral peptides using tri-segment evolutionary profile and word embedding based multi-perspective features with deep stacking model. BMC Bioinformatics. 2024;25(1):102. doi:doi.org/10.1186/S12859-024-05726-5.

33. Akbar S, Zou Q, Raza A, Alarfaj FK. iAFPs-Mv-BiTCN: Predicting antifungal peptides using self-attention transformer embedding and transform evolutionary based multi-view features with bidirectional temporal convolutional networks. Artificial Intelligence in Medicine. 2024;151:102860. doi:doi.org/10.1016/j.artmed.2024.102860.

34. Akbar S, Ali F, Hayat M, Ahmad A, Khan S, Gul S. Prediction of Antiviral peptides using transform evolutionary & SHAP analysis based descriptors by incorporation with ensemble learning strategy. Chemometrics and Intelligent Laboratory Systems. 2022;230:104682. doi:doi.org/10.1016/j.chemolab.2022.104682.

35. Akbar S, Hayat M, Iqbal M, Jan MA. iACP-GAEnsC: Evolutionary genetic algorithm based ensemble classification of anticancer peptides by utilizing hybrid feature space. Artificial Intelligence in Medicine. 2017;79:62–70. doi:doi.org/10.1016/j.artmed.2017.06.008.

36. Akbar S, Rahman AU, Hayat M, Sohail M. cACP: Classifying anticancer peptides using discriminative intelligent model via Chou’s 5-step rules and general pseudo components. Chemometrics and Intelligent Laboratory Systems. 2020;196:103912. doi:doi.org/10.1016/j.chemolab.2019.103912.

37. Ullah M, Akbar S, Raza A, Zou Q. DeepAVP-TPPred: identification of antiviral peptides using transformed image-based localized descriptors and binary tree growth algorithm. Bioinformatics. 2024;40(5):btae305. doi:doi.org/10.1093/bioinformatics/btae305.

Additionally, we have conducted a thorough review of all other references in the manuscript. Upon careful examination, we have ensured that all citations are correctly formatted and accurately reflect the sources. We deeply regret the oversight in the manuscript and have taken extra care to avoid such errors moving forward.

We sincerely appreciate your attention to detail and your guidance, which has greatly improved the accuracy and quality of our manuscript. We look forward to your further assessment and hope that our revisions meet your expectations.

We apologize once again for the oversight in our previous submission, and we greatly appreciate your careful attention in pointing out this issue.

Thank you once more for your valuable feedback and guidance.

---

## [Editor Report · Decision Letter 2]

2 Oct 2024

Predicting Learning Achievement Using Ensemble Learning with Result Explanation

PONE-D-24-26784R2

Dear Dr. Tong,

We’re pleased to inform you that your manuscript has been judged scientifically suitable for publication and will be formally accepted for publication once it meets all outstanding technical requirements.

Kind regards,

Shahid Akbar, PhD

Academic Editor

PLOS ONE
---

## [Editor Report · Acceptance letter]

10 Oct 2024

PONE-D-24-26784R2 

PLOS ONE

Dear Dr. Tong, 

I'm pleased to inform you that your manuscript has been deemed suitable for publication in PLOS ONE. Congratulations! Your manuscript is now being handed over to our production team.

Kind regards, 

on behalf of

Dr. Shahid Akbar 

Academic Editor

PLOS ONE